# Sugar Content of Market Beverages and Children’s Sugar Intake from Beverages in Beijing, China

**DOI:** 10.3390/nu13124297

**Published:** 2021-11-28

**Authors:** Jing Wen, Huijuan Ma, Yingjie Yu, Xiaoxuan Zhang, Dandan Guo, Xueqian Yin, Xiaohui Yu, Ning Yin, Junbo Wang, Yao Zhao

**Affiliations:** 1Department of Nutrition and Food Hygiene, School of Public Health, Peking University, Beijing 100191, China; wenjing@bjmu.edu.cn (J.W.); mhjmhjmhj717@163.com (H.M.); zxxyingyang@163.com (X.Z.); yinxq@pku.edu.cn (X.Y.); yinning@bjmu.edu.cn (N.Y.); 2Beijing Center for Disease Prevention and Control, Beijing Research Center for Preventive Medicine, Beijing 100013, China; yyj.jane.1982@163.com (Y.Y.); aldblm.tm@163.com (D.G.); yxh770770@sina.com (X.Y.); 3Beijing Key Laboratory of Toxicological Research and Risk Assessment for Food Safety, Peking University, Beijing 100191, China

**Keywords:** sugar, beverages, children, consumption

## Abstract

(1) Background: This study aims to find the sugar content of market beverages and estimate the sugar intake from beverages among students in Beijing. (2) Methods: Using snapshotting, we collected the sugar content of beverages through their packages or nutrition labels. Combined with the statistic of student beverage consumption, we estimated students’ sugar intake. (3) Results: The median sugar content of total beverages was 9.0 g/100 mL, among which the fruits/vegetable juices and beverages had the highest sugar content (10.0 g/100 mL). Sugar content in most beverages in Beijing was generally higher than the recommendations, and fruit/vegetable juices and beverages exceeded the most. The median of sugar intake from beverages among students was 5.3 g/d, and the main sources were fruit/vegetable juices and beverages, protein beverages and carbonated beverages. Sugar intake from beverages differed according to gender, age and living area. Higher sugar intake was found among boys, older students and rural students. (4) Conclusions: Sugar content in market beverages in Beijing were high. Gender, age and residence were the influencing factors of sugar intake. Targeted measures should be taken to decrease the sugar content in beverages, especially the fruit/vegetable juices and beverages and the sugar intake among students.

## 1. Introduction

Free sugar, including monosaccharides and disaccharides added to foods by food manufacturers or processors, is found mostly in sugar-sweetened beverages (SSBs) [1]. SSBs are defined as sugar-sweetened sodas, fruit drinks, sweetened tea beverages, sports/energy drinks and other drinks with added sugar [2]. Youths were reported to consume more SSBs than adults or elders [3]. Frequent intake of excessive SSBs is strongly associated with obesity [4], diabetes [5], hypertension [6] and dental caries [7] among children due to the high sugar and energy content and little nutritional value of SSBs. With the rapid urbanization, economic improvement and increasing westernized diet in China, there were 66.6% of the children consuming SSBs in 2013 [8], and the proportion of children who consumed SSBs more than once per week increased greatly from 14.2% in 2002 [9] to 79.2% in 2012 [10].

Therefore, policies and guidelines were intended to address a surge in non-communicable diseases (NCDs) with SSBs, recommending to drink less or avoid SSBs. For the United Kingdom (the UK) and French government [11,12], the guidance on front-of-pack (FOP), color-coded labeling was to help consumers to recognize the level of sugar content of the product in different colors. Sugar tax was increased first in Hungary [13], Finland [14] and some other European countries [15]. In China, the maximum of 50 g and best under 25 g added sugar per day was suggested in the dietary guidelines [16], which was also emphasized in the National Nutrition Program (2017–2030).

According to the Chinese General Standard for Beverage (GB/T 10789-2015) [17], which was released in 2015 and came into effect in 2016, the common beverages on sale are divided into eight categories: protein beverages, fruit/vegetable juices and beverages, carbonated beverages, beverages for special uses, flavored beverages, tea beverages, coffee beverages and botanical beverages (the detailed definition is shown in Appendix A). However, previous studies in China focused more on the consumption frequency and consumption quantity of beverages among children. Although a few studies revealed the sugar and energy content of carbonated [18] and non-carbonated beverages [19] (including juice drinks, tea-based beverages, sports drinks and energy drinks) in Beijing, the sugar content of other common beverages, such as protein beverages, were still unclear, and the sample sizes were not big enough (93 carbonated beverages and 463 non-carbonated beverages). Besides, only 1 study took the quantity of sugar intake of children from beverages into account [20]. For this reason, it is our duty to focus on the sugar content in each beverage, the amount of sugar children took from the beverages each time, and whether the amount met the recommending standards or not under the current open policy.

Thus, in order to investigate the sugar content of market beverages and how much sugar students could get daily from beverages, we combined the statistic of a cross-sectional study with a consumption survey. A cross-sectional study with a sample of 26 supermarkets in Beijing was performed during October to December 2019, which aimed to investigate the sugar content of beverages. Besides, aiming to investigate the beverages consumption among students in Beijing, we also selected a portion of the data from 2019 Students Snacks and Beverages Consumption Survey in Beijing for a secondary data analysis.

## 2. Materials and Methods

This study could be divided into 2 parts: a cross-sectional study about the sugar content of beverages for sale and a secondary data analysis about children’s beverage consumption.

### 2.1. Cross-Sectional Study

#### 2.1.1. Selection of Sampling Site

We conducted a cross-sectional study in supermarkets in Beijing from October to December 2019. The selection of sampling sites mainly comprised three steps: Firstly, based on the Kantar World Panel Research [21], 16 supermarket chains of 12 retail groups in the top 10 Grocery Market Shares of China, Northern China and Eastern China were included in our study, excluding those without branches in Beijing. Secondly, we included supermarket chains with more than 4 branches in Beijing, expanding our study spots into 20 supermarket chains. Thirdly, considering that the convenience stores such as Lawson, 7-Eleven and so on were also common beverage consumption sites, we added 6 convenience stores into our selection of samples. Data of supermarket chains in Beijing, including name, total number of branches and addresses of each supermarket chain, were captured from the free web-based, geospatial information provider Amap.

In all, 26 supermarket chains or convenience stores were chosen in this study (RT-Mart (Taiwan, China), Auchan, Vanguard, Ole, Wal-Mart (Bentonville, AR, USA), Yonghui Superstores (Fuzhou, China), Carrefour, Hualian, Kuaike, Wu Mart (Beijing, China), Merry Mart, Huaguan, Dashang Supermarket, Jingkelong Supermarket (Beijing, China), Inzone, Lotus Market, Cuiwei Department Store (Beijing, China), Xingfurongyao Supermarket, CSF Market, Shijijiajiafu Supermarket, Lawson (Shinagawa, Japan), 7-Eleven (Dallas, TX, USA), Good Neighbor, Family Mart, BianLiFeng and Our Hours). For each supermarket chain, only 1 supermarket was visited in our study due to the high internal similarity in suppliers.

#### 2.1.2. Definition and Selection of Beverages

Our beverages were divided into 8 categories in accordance with the Chinese General Standard for Beverage. Beverages without carbohydrate/sugar/energy content on their nutrition labels and beverages without nutrition labels were excluded from this study. The definition of imported and domestic products was based on the place of origin which was printed on the product package. The imported products referred to those beverages who labeled on the package that the country of origin was a foreign country. Those beverages without the country of origin and the producing area in China were defined as domestic products.

#### 2.1.3. Data Collection of the Beverages

Data were collected from product packaging and nutrient information panels of beverages by snapshotting all the beverages for sale which met the study inclusion criteria, to obtain the company name, product name, origin, package size, carbohydrate, sugar and energy content per 100 mL of each kind of beverage. For beverages without sugar content on the package, we used carbohydrate content as a replacement, and for those beverages with same formulation but different in size, we only recorded once. Besides, as the number of supermarkets visited increased, we only recorded products that had not been photographed in previous supermarkets in the subsequent study sites.

#### 2.1.4. The Criteria for Free Sugar Intakes

In this study, we used 3 kinds of recommendations for free sugar intakes to evaluate the sugar content in different categories of beverages: (1) The guideline released by the UK Food Standards Agency (FSA) on FOP color-coded labeling for drinks, which was also called ‘traffic light system’ (red, high levels; amber, medium levels; green, low levels). Cutoffs for the sugar content criteria per 100 mL in this guideline were as follows: high >11.25 g, medium >2.5 g and ≤11.25 g, low ≤2.5 g. Cutoff of high sugar content criterion per portion was >13.5 g. (2) The recommendation by the World Health Organization (WHO) on the restriction of 25 g daily free sugar intake, which would reduce the risk of non-communicable diseases. (3) The recommendation by the Dietary Guideline for Chinese Residents on the restriction of 50 g added sugar, and best not more than 25 g per day, for otherwise it would increase the risk of dental caries and overweight.

### 2.2. Secondary Data Analysis

#### 2.2.1. Study Population

The secondary data were derived from the result of 2019 Students Snacks and Beverages Consumption Survey in Beijing. After choosing 7 districts in Beijing (Xicheng District, Chaoyang District, Haidian District, Tongzhou District, Changping District, Fangshan District and Miyun District), 4 primary schools and 4 middle schools were selected from each district for investigation. Eligible full-time students in the first, third, fifth and seventh grade without severe birth defects and severe diseases and one of their parents were enrolled in the survey. Students were recruited directly in school by investigators. The survey was conducted via questionnaire in all recruited students and adults to investigate the consumption of snacks and beverages of students and related family environment. Written informed consent was obtained from the students and their parents, and all information was kept confidential. This study was approved by the medical ethics committee of the Beijing Center for Disease Control and Prevention (approval number: No. 14, 2019).

#### 2.2.2. Variable Selection

The 2019 Students Snacks and Beverages Consumption Survey in Beijing could be divided into 5 domains: (1) students’ general information; (2) students’ knowledge and attitude about the snacks and beverages; (3) general information on the main caregiver of the students; (4) students’ consumption of snacks and beverages in the past week; (5) family snacks and beverages environment. Considering the younger age of first-graders, the survey of their snack and beverage consumption was completed by their parents and included in their parents’ questionnaire.

A total of 3 items (gender, age and residence) in students’ general information domain and a total of 18 items (consumption frequency and quantity of milk drink, carbonated beverages, pure fruit/vegetable juice, fruit/vegetable beverages, tea beverages, beverages for special uses, soya-bean milk, protein beverages and other beverages) in students’ beverage consumption in the past week were included in our study. According to the Chinese General Standard for Beverage, we merged several variables such as “milk drinks”, “soya-bean milk” and “protein beverages” into “protein beverages”. “Pure fruit/vegetable juice” and “fruit/vegetable beverages” were combined into “fruit/vegetable juices and beverage”. Altogether, 5 categories of beverages (protein beverages, fruit/vegetable juices and beverage, carbonated beverages, tea beverages and beverages for special uses) were included in our study. The consumption frequency and single consumption quantity of 5 categories of beverages were used as dependent variables, while the demographic was used as independent variable.

#### 2.2.3. Variable Conversion

Due to the current study’s method and analysis, part of original variables from the raw data were converted. Ages of students were calculated by students’ birthdates and were categorized as “6–8 years old”, “8–10 years old”, “10–12 years old” and “12–14 years old”. In addition, the beverage consumption rate was the ratio of the number of students who had consumed any beverage in the past week to the number of students participating in the survey. Consumption frequency was calculated by adding the numbers of consumption frequency of the 5 types of beverages in the past week. Consumption quantity referred to the average daily beverage consumption, which was estimated with the formula:(1)$$\mathrm{Consumption}\;\mathrm{quantity}\;\left(\mathrm{mL}/\mathrm{day}\right)=\frac{\mathrm{Consumption}\;\mathrm{frequency}\;\mathrm{in}\;\mathrm{the}\;\mathrm{past}\;\mathrm{week}\;\times \;\mathrm{Average}\;\mathrm{beveragesconsumption}\;\mathrm{quantity}}{7}$$

The sugar intake from beverages could be evaluated by the following formula:(2)$$\mathrm{Sugar}\;\mathrm{intake}\;\mathrm{of}\;\mathrm{each}\;\mathrm{beverage}\;\left(g/d\right)=\mathrm{Average}\;\mathrm{sugar}\;\mathrm{content}\;\mathrm{of}\;\mathrm{each}\;\mathrm{beverage}\times \mathrm{Consumption}\;\mathrm{quantity}$$

According to the recommendation of daily sugar intake from the Dietary Guideline for Chinese Residents, students were separated into low-consumption group (<25 g/d), middle-consumption group (25–50 g/d) and high-consumption group (>50 g/d).

### 2.3. Statistical Analysis

Epidata3.1 software (The Epi Data Association, Odense, Denmark) was used for data entry, and all data received were double-checked. All statistical analyses were conducted via IBM SPSS V.23.0 software (IBM, Aromonk, NY, USA). Due to the non-normal distribution, description information was presented as median, interquartile ranges (IQR), numbers (n) or proportions (%). For categorical and continuous variables, the analysis of group differences in sugar content, students’ beverage consumption, students’ sugar intake data and sugar intake level were conducted by Chi-square test and non-parametric Kruskal–Wallis test. McNemar–Bowker test was performed to compare of the proportion of the red-light products per 100 mL criterion and per portion criterion according to the FSA traffic lights system. Spearman’s rank correlation coefficient and related-samples Wilcoxon signed-rank test were applied to explore the correlation between sugar and carbohydrate in beverages with both contents shown on their packages. Ordinal logistic regression model was performed to analyze the factors affecting the sugar consumption level among students. Two-side *p* values < 0.05 were considered statistically significant.

## 3. Results

### 3.1. Sugar and Energy Content in Beverages

Overall, 1587 beverages met the inclusion criteria for the statistical analysis. Based on the GB/T 10789-2015, all the products were classified into eight categories, with 39.8% fruit/vegetable juices and beverages, 16.6% protein beverages, 12.9% carbonated beverages, 10.4% tea beverages, 7.6% flavored beverages, 6.9% coffee beverages, 3.2% beverages for special uses and 2.3% botanical beverages. According to the origin places, imported products accounted for 38.6% of the total beverages and domestic products accounted for 61.4%.

Table 1 shows the sugar and energy content in different beverages. Generally, the median of the package size in all products was 350 (IQR: 250–500) mL. The median of sugar and energy content was 9.0 (IQR: 6.0–11.0) g/100 mL or 29.7 (IQR: 17.5–45.5) g/serving and 42.1(IQR: 28.5–50.5) kcal/100 mL or 140.0 (IQR: 87.3–215.2) kcal/serving, respectively. Statistical differences were evaluated in the sugar and energy content both per 100 mL and per serving among different categories of beverages (*p* < 0.001). Multiple comparison tests revealed fruit/vegetable juices and beverages had the highest sugar content per 100 mL among all products, and sports drinks had the lowest sugar and energy content per 100 mL. Besides, imported products had significantly higher levels of both sugar and energy content than domestic beverages (the median of sugar content: 10.0 g/100 mL vs. 8.0 g/100 mL, *p* < 0.001; the median of energy content: 45.2 kcal/100 mL vs. 39.2 kcal/100 mL, *p* < 0.001).

### 3.2. Portions of Sugar Exceeding Rate with Different Criteria

On the basis of different criteria for added sugar intake, the sugar content in different categories of beverages is summarized in Table 2. According to the FSA, 344 (21.7%) beverages were marked with a red label in the per 100 mL criterion, while 1309 (82.5%) products were red in the per serving criterion. Meanwhile, 955 (60.2%) products had an excessive sugar content exceeding the daily free sugar intake recommendation from WHO. Among total 1587 products, the McNemar−Bowker test showed statistically significant results on the difference in the proportion of products marked with a ‘red’ label when using the per 100 mL standard and per serving standard (χ2 = 933.966, *p* < 0.001). Based on the different criteria, juice drinks had the highest percentage of products (33.6%, 96.8%, 82.3%) exceeding the UK’s per 100 mL, per serving and WHO criterion, while coffee beverages had the lowest percentage of products (0.9%, 67.3%, 21.8%).

Among 1587 beverages, we only found sugar content on the nutrition labels on 298 (18.8%) products. Carbonated beverages had the highest proportion of products with sugar labeling (34.1%), while protein beverages had the lowest (8.7%). No statistical difference was found between the imported and domestic products regarding the sugar labeling. For those beverages with both sugar and carbohydrate content information, correlations of these 2 contents were calculated by Spearman’s rank correlation coefficients and showed a strong correlation (*r* = 0.964, *p* < 0.001, Appendix B, Table A2). The differences between carbohydrate and sugar contents were significant (Wilcoxon matched pair test, *p* < 0.001, Appendix B, Table A3).

### 3.3. Beverage Consumption among Students

Of the total number of 3957 students, results from 3908 students were analyzed excluding 49 students due to serious data loss (≥70%). While 50.4% were boys, students in this survey aged from 6 to 14 (Table 3). Among these, the proportions of children of all ages were close to 25%, and more than half of the students (58.4%) lived in rural areas. In total, 83.2% of the students consumed SSBs in the past week (83.4% for boys and 83.0% for girls). Students aged over 10 presented a higher SSBs consumption rate than those aged under 8. Besides, students residing in rural areas consumed more SSBs than those urban students. (*p* < 0.001). The consumption rate, frequency and single consumption quantity of each kind of beverage are shown in Appendix C, Table A4, Table A5 and Table A6.

The total median of consumption frequency and single consumption quantity were 3.0 (IQR: 1.0–7.0) per week and 63.1 (IQR: 11.9–182.7) m/d. Boys drank more than girls in every time they consumed (*p* = 0.039). Significant differences existed in consumption frequency and single consumption quantity between students with different age and residence (*p* < 0.001).

### 3.4. Sugar Intake from Beverages among Students

Combining the data in Table 1 and Table 3, we estimated the sugar intake from each kind of beverage (Table 4). Overall, the median of the sugar intake from beverages was 5.3 (IQR: 1.0–15.2) g/d. Boys, students over 8 years old and rural students presented higher sugar intake than girls, students under 8 years old and urban students (*p* < 0.05).

### 3.5. Sugar Consumption Levels among Students

Among all the students in our study, 85.1% of the students’ sugar intake from beverages was lower than 25 g/d, and 5.9% of the students consumed excess added sugar from the beverages which was higher than the maximum level of the recommendation (Table 5). More girls were found in the low-consumption group, and more boys were in middle-consumption group (*p* < 0.05). There was no statistically significant difference in the high-consumption groups between boys and girls. Older students and rural students were associated with higher sugar consumption (*p*_all_ < 0.001).

### 3.6. Factors Affecting Consumption Levels among Students

An ordinal logistic regression model was conducted to explore the influencing factor of students’ sugar consumption levels (Model Fitting Information: *p* < 0.001; Test of Parallel Lines: Chi-Square: 3.676, *p* = 0.597). The results are shown in Table 6. Our results found that boys consumed more sugar from beverages significantly more than girls (OR = 1.309, 95% CI: 1.091~1.570, *p* = 0.004). Compared with students aged 12–14, students who were under 10 years old consumed less sugar from beverages (6–8 years old: OR = 0.087, 95% CI: 0.057–0.133, *p* < 0.001; 8–10 years old: OR = 0.645, 95% CI: 0.513–0.811, *p* < 0.001). There was no difference between the students aged 10–12 and students aged 12–14 in the sugar consumption level. For residence, students living in urban areas consumed less sugar than the rural students (OR = 0.449, 95% CI: 0.368–0.548, *p* < 0.001).

### 3.7. Proportions of Sugar Intake from Each Kind of Beverage in Total Sugar Intake from Beverages

As shown in Figure 1, the top three sources of sugar intake from beverages were protein beverages (38.15%), fruit/vegetable juices and beverages (24.31%) and carbonated beverages (15.01%). For gender, the proportions of girls’ sugar intake from protein and fruit/vegetable juices and beverages were significantly larger than those of boys, but the proportions of carbonated beverages and beverages for special uses were less than boys (*p*_all_ < 0.05). For students who were 6–8 years old, the proportions of sugar intake from protein, carbonated, tea beverages and beverages for special uses were less than students of other ages (*p*_all_ < 0.001), and the proportions of fruit/vegetable juices and beverage was only less than the proportions of those students aged 8–10. For those students aged 12–14, the proportions of sugar intake from these five kinds of beverages were larger than students who were 8–10 years old. For the rural students, the proportions of sugar intake from carbonated, tea beverages and beverages for special uses were significantly larger than urban students’ (*p*_all_ < 0.001). Compared with the rural students, although the proportions of protein beverages and fruit/vegetable juices and beverages were higher among urban students, significant differences were not found.

## 4. Discussion

As far as we know, previous studies had limitations on sampling methods, survey districts and smaller sample sizes, whereas our study established the first Beverages Nutrition Labels Database in Beijing, which covered 1587 beverages among 26 supermarket chains or stores. Besides, combining the nutrition labels database with the data of students’ beverage consumption, our study is the first study to estimate students’ sugar intake from beverages in Beijing.

In our study, the median of sugar content of total beverages was 9.0 g/100 mL, which was lower than the results from western countries such as the United States (the US, 11.3 g/100 mL) [22] and Canada (12.5 ± 19.5 g/100 mL) [23]. The observation was echoed with the finding that the sugar content of imported products was significantly higher than the locals’ [24]. The median sugar content of per 100 mL fruit/vegetable juices and beverage, protein beverages and carbonated beverages in our study was 10.0 g, 8.0 g and 8.0 g, respectively. The sugar contents were different to the result in the UK (carbonated SSBs: 30.1 ± 10.7 g/330 mL) [25] and the results in New Zealand (fruit juices: 9.8 ± 3.6 g/100 mL; dairy/soy-based drinks: 8.2 ± 1.7 g/100 mL; carbonated drinks: 8.7 ± 5.5 g/100 mL) [26]. We also found the sugar content in present study was similar to the previous studies in China (carbonated beverages: 9.3 g/100 mL [18], non-carbonated beverages: 9.6 g/100 mL) [19]. Differences in sample sizes, products formula, manufacturers, flavors and brands might account for these results. For those beverages without sugar content on their nutrition label, Wilcoxon matched pair test showed that the carbohydrate was more than sugar and could not represent the sugar content well, including fruit/vegetable juices and beverage, protein beverages, flavored beverages, tea beverages and coffee beverages. Sucrose, fructose and glucose in fruits or vegetables [27], lactose, starch [28] in plant-based milk beverages and polysaccharides from coffee beans [29] remaining in the coffees might lead to these results.

High sugar content in beverages is usually in relation to high sugar exceeding standard rate. The WHO recommendation (added sugar intake below 25 g/d) was only met by 39.8% of beverages in this study, which was higher than the results in carbonated (37.6%) [18] and non-carbonated (18.4%) [19] beverages in Beijing. Trying to fully incorporate all beverages (while including lower-sugar and sugar-free beverages) might contribute to the rise of the qualified sugar content rate. Fruit/vegetable juices and beverages enjoyed the highest sugar exceeding rate, which was not only according to the UK criteria (per 100 mL: 33.6%; per package size: 96.8%) but the recommendation from WHO (82.3%) as well. The result appeared to be consistent with the discovery from the previous study of juice drinks in Beijing in 2018 (criteria from the UK per 100 mL: 32.2%, per serving portion: 94.1%; WHO recommendation: 83.0%) [19].

Through the significant difference between the results according to the two kinds of criteria from FSA, our study confirmed that package size is a major influencing factor of the sugar content in total beverages, for a much higher exceeding rate was shown when converting sugar content from per 100 mL to per package size (21.7% vs. 82.5%). In general, larger package size contributes to a larger consumption by consumers [30] and results in excessive energy intake [31], even weight gain and obesity [32]. Therefore, in the present situation of high sugar content, sugar exceeding rate and large package size in beverages, the problem of high free sugar in beverages which represented a global challenge in public health should be taken seriously.

On the other hand, the consumption of beverages is closely linked with the added sugar intake. Integrated with the sugar content from nutrition labels and the beverages consumption among students in Beijing, we roughly evaluated students’ sugar intake in Beijing in 2019. The median of sugar intake from beverages among students was 5.3 g/d, which tended to be much lower than the sugar intake in other countries. The Canadian research revealed the mean sugar intake from beverage was 47 g/d among 9–18-year-old children in 2015 [33], and a related Korean study showed the daily total sugar intake from beverages was 13.76 g [34]. Differences in the preparation process of the beverages, lower sugar content in domestic products and lower beverages consumption in students might be the reasons for the discrepancy in results. According to the suggestion on free sugar intake from WHO, 14.9% of the students in our study were likely to be less adherent to the advice. 16.4% of the boys and 18.6% of the students living in rural areas consumed excessive sugar. The differences in the proportions of consuming sugar over 25 g/d were significant between the students under 8 years old and over 8 years old, for the number of students aged over 8 who exceed the sugar limit was much higher than that of students aged under 8. All of these factors can probably be explained by with the larger beverage size of each intake and higher frequency of the beverage intake. These findings were also reported by a previous study [35]. Greater demand of activities and energy among boys and older students, more discretionary pocket money, greater freedom of snack choice among older students and the nutrition education gap between urban and rural areas may cause the differences in beverage consumption. Parent- and school-based intervention can strongly influence the food choices of students [36,37]. Parents and teachers are advised to provide targeted nutrition education to students to reduce beverage consumption, especially for boys, older students and students in rural areas.

We also found the top three contributors to beverage consumption and sugar intake were protein beverages, fruit/vegetable juices and beverage and carbonated beverages. A study in China found the similar result that juice and milk beverages were more popular than carbonated and tea beverages among Chinese children [10]. Protein beverages in our study were not the beverages with the most sugar content, but it was the most popular beverage among students, which indicated the intake of protein beverages was the highest. This suggested that students or parents highly valued the protein beverages, probably with the knowledge that milk can help children a lot during their growth [38,39], and the improvement of the public’s health awareness. However, protein beverages are different from milk, and they cannot replace milk. Consumers may be misled by advertisement which exaggeratedly claims the health benefits of these products. Sugar from fruit/vegetable juices and beverages occupied about 24.31% of the total sugar intake from beverages in present study. Besides, the high level of natural sugars in fruit/vegetable juices cannot be ignored, and the long-term, frequent, excessive intake of juices is associated with weight gain [40], type 2 diabetes [4] and metabolic syndrome [41].

Overall, the questions of high sugar content in beverages and high consumption of fruit/vegetable juices and beverages among students could not be ignored. Therefore, all parties are supposed to take effective measures to strengthen the control of reducing sugar intake in beverages, such as restricting package size, improving nutrition labels management and public health awareness and so on. The government should take measures to restrain the sugar content in beverages, especially the imported beverages and fruit/vegetable juices and beverages. Restricting the package size of beverages is an effective method, and it has been confirmed by previous studies [42,43]. A simulated cost-effectiveness study from Australia showed that the 375 mL package limit of SSBs tended to reduce energy intake, weight and BMI and lighten economic burden [44]. Policies on the limitation of beverage package sizes have been published in the US [45].

With the advantages of reliability and accessibility to the nutrition information, nutrition labels can guide consumers to a better choice. Among 1587 beverages in our study, only 298 (18.8%) products had sugar content on their nutrition label, probably with the reason that sugar content is not compulsively required on the nutrition label in China [46]; therefore, the improvement of nutrition label standards is essential. In the European countries, the FOP traffic lights system among prepackaged food was recommended to help consumers identify healthier food, and it was most widely applied in the UK [47]. In the US, the added sugar content on nutrition labels is stipulated by the Food and Drug Administration (FDA) [48], and health warnings on the packages of beverages were demanded in some states [49,50].

In addition, in the inherent impression of consumers of the benefits of fruits and vegetables leads to the stereotype that fruit/vegetable juices and beverages are healthy and can be consumed without worries. However, the health concern from the excessive fruit/vegetable juices and beverages should not be neglected. The department of education and the media should help consumers to raise awareness on the sugar content and the adverse effects of beverages, especially the fruit/vegetable juices and beverages. Meanwhile, teachers or parents should guide the students correctly to reduce their beverage consumption.

There were some limitations that should be noted in our study. Firstly, although beverages are also available online or in other smaller retail stores, we visited 26 supermarkets in Beijing including supermarket chains from top 10 retail groups, local supermarket chains and convenience stores, whose beverages could represent the common selling beverages well. Secondly, assessing daily beverages intake from the 2019 Students Beverages and Snacks Consumption Survey in Beijing might be affected by recall bias and could not accurately reflect students’ daily beverages intake. Thirdly, the beverages without nutrition labels were not included in our study. Therefore, further studies are required to detect the sugar content in those products through professional methods. Fourthly, due to the lack of students’ dietary data, the contribution rates of the sugar and energy from beverages to the dietary sugar and energy are unknown in our study, which can be improved in further studies.

## 5. Conclusions

The sugar content of beverages was high in China, and fruit/vegetable juices and beverages had the most sugar. Children’s sugar intake from beverages in China was lower than developed countries. The sugar from beverages mostly came from protein beverages, fruit/vegetable juices and beverages and carbonated beverages. Higher sugar intake was found among boys, students over 10 years old and rural students. Strategies including reducing sugar content, restricting the package size of beverages, enhancing the management of nutrition labels and properly guiding the public understanding of beverages, especially fruit/vegetable juices and beverages, should be taken to decrease sugar intakes in China.

## Figures and Tables

**Figure 1 nutrients-13-04297-f001:**
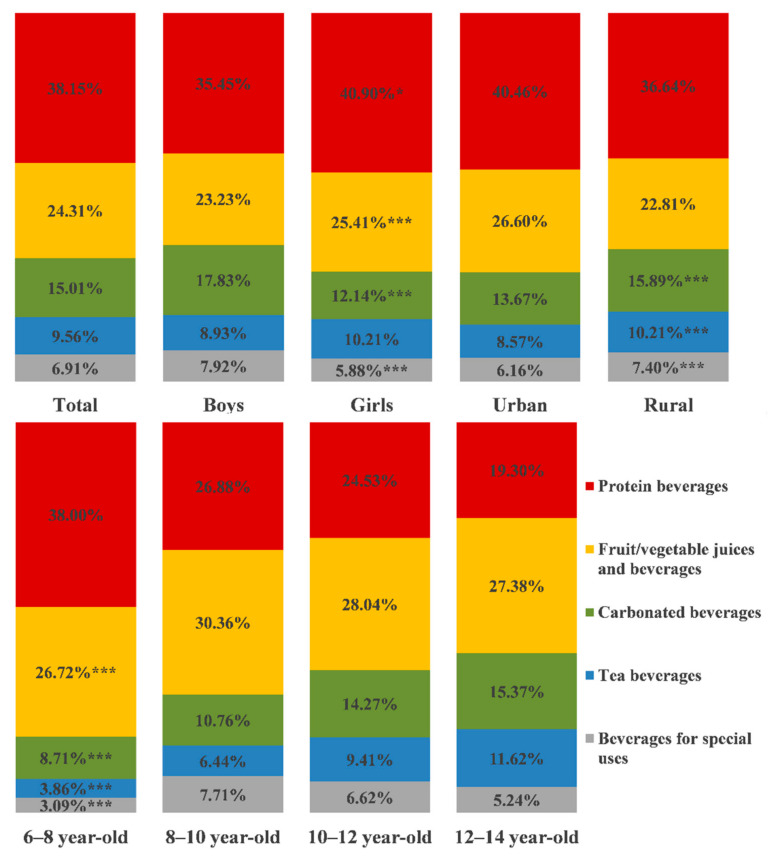
Proportions of sugar intake from each kind of beverage in total sugar intake from beverages. Sources of sugar intake from each beverage by gender, age and residence (the proportion of total sugar intake from beverages, %). * *p* < 0.5, *** *p* < 0.001. Compared with by Mann–Whitney U test and Kruskal–Wallis test in non-parametric test. Detailed differences between groups are shown in Appendix D, Table A7.

**Table 1 nutrients-13-04297-t001:** Description of median (IQR) sugar and energy content in different beverages.

Categories	*n* (%)	Package Size (mL)	Sugar Content (g) Median (Range)		Energy Content (kcal) Median (Range)	
		Median (Range)	Per 100 mL	Per Serving	Per 100 mL	Per Serving
Total	1587 (100)	350 (250–500)	9.0 (6.0–11.0)	42.1 (28.5–50.5)	29.7 (17.5–45.5)	140.0 (87.3–215.2)
Fruit/vegetable juices and beverages	631 (39.8)	450 (290–1000)	10.0 (8.0–12.0)	44.5 (37.3–50.0)	40.6 (28.1–110.0)	173.2 (121.8–449.6)
Protein beverages	264 (16.6)	250 (200–340)	8.0 (6.0–11.0)	50.0 (41.9–62.7)	19.8 (12.5–30.6)	130.9 (89.4–202.3)
Carbonated beverages	205 (12.9)	340 (330–500)	8.0 (5.0–11.0)	34.9 (19.6–44.5)	28.7 (16.5–42.0)	116.7 (65.0–170.9)
Tea beverages	165 (10.4)	500 (385–500)	7.0 (0–9.0)	32.5 (5.5–43.6)	27.3 (0–40.0)	143.5 (24.4–185.5)
Flavored beverages	121 (7.6)	360 (315–495)	6.0 (3.0–10.0)	23.2 (13.0–45.7)	22.0 (8.0–30.3)	92.1 (38.6–126.2)
Coffee beverages	110 (6.9)	268 (240–281)	7.5 (3.0–8.0)	44.7 (18.6–56.3)	19.2 (8.0–23.1)	118.1 (47.4–159.3)
Beverages for special uses	50 (3.2)	500 (420–600)	5.0 (4.0–6.0)	25.0 (21.0–29.2)	30.0 (20.0–33.0)	129.1 (108.8–146.3)
Botanical beverages	41 (2.3)	310 (300–450)	8.0 (4.0–9.0)	31.6 (22.0–37.1)	21.7 (13.4–30.1)	96.4 (69.9–129.9)
*p*		<0.001	<0.001	<0.001	<0.001	<0.001
Imported products	612 (38.6)	342.5 (250–1000)	10.0 (7.0–12.0)	45.2 (34.9–52.8)	33.0 (20.0–70.0)	147.0 (99.0–299.8)
Domestic products	975 (61.4)	350 (268–500)	8.0 (5.0–10.0)	39.2 (25.6–48.5)	27.5 (16.5–40.0)	133.3 (83.8–187.7)
*p*		<0.001	<0.001	<0.001	<0.001	<0.001

**Table 2 nutrients-13-04297-t002:** Description of sugar in different beverages according to different recommendations (*n*, %).

Categories	Number of Products with ‘Red’ Label Color-Coded Labeling		Number of Products with Free Sugar > WHO Recommendation (>25 g)	Number of Products with Free Sugar > Chinese Dietary Guideline Recommendation (>50 g)
	>11.25 g/100 mL	>13.5 g/Serving		
Total	344 (21.7)	1309 (82.5)	955 (60.2)	372 (23.4)
Fruit/vegetable juices and beverages	212 (33.6)	611 (96.8)	519 (82.3)	271 (42.9)
Protein beverages	58 (22.0)	187 (70.8)	100(37.9)	31 (11.7)
Carbonated beverages	34 (16.6)	162 (79.0)	119 (58.0)	39 (19.0)
Tea beverages	22 (18.2)	83 (68.6)	54 (44.6)	11 (9.1)
Flavored beverages	12 (7.3)	117 (70.9)	91 (55.2)	13 (7.9)
Coffee beverages	1 (0.9)	74 (67.3)	24 (21.8)	0 (0)
Beverages for special uses	3 (6.0)	44 (88.0)	34 (68.0)	2 (4.0)
Botanical beverages	2 (4.9)	31 (75.6)	14 (34.1)	5 (12.2)
*p*	<0.001	<0.001	<0.001	<0.001
Imported products	263 (21.2)	122 (43.9)	159 (25.2)	195 (31.9)
Domestic products	35 (10.2)	176 (13.4)	139 (14.6)	177 (18.2)
*p*	<0.001	<0.001	<0.001	<0.001

**Table 3 nutrients-13-04297-t003:** Consumption rate, frequency and single consumption quantity among students.

Categories		*n* (%)	Consumption Rate (%)	*p*	Consumption Frequency P50 (IQR) (Times/Week)	*p*	Single Consumption Quantity P50(IQR) (mL/d)	*p*
Total		3908 (100)	83.2		3.0 (1.0–7.0)		63.1 (11.9–182.7)	
Gender	Boys	1971 (50.4)	83.4	0.709	3.0 (1.0–7.0)	0.815	71.4 (14.3–195.2)	0.039
	Girls	1937 (49.6)	83.0		3.0 (1.0–7.0)		57.1 (10.1–171.4)	
Age	6–8	1010 (25.8)	78.0	<0.001	2.0 (1.0–4.0)	<0.001	21.4 (0–69.0)	<0.001
	8–10	974 (24.9)	83.8		3.0 (1.0–8.0)		71.4 (13.7–196.8)	
	10–12	937 (24.0)	86.9		3.0 (1.0–8.0)		89.3 (28.6–228.4)	
	12–14	985 (25.2)	84.5		4.0 (1.0–7.0)		102.4 (25.4–266.7)	
Residence	Urban	1627 (41.6)	79.9	<0.001	3.0 (1.0–5.0)	<0.001	42.9 (4.8–131.0)	<0.001
	Rural	2281 (58.4)	85.5		3.0 (1.0–8.0)		82.1 (15.7–219.2)	

**Table 4 nutrients-13-04297-t004:** Sugar intake from beverages in different groups of students (g/d, P50 (IQR)).

Categories		All Beverages	Fruit/Vegetable Juices and Beverages	Protein Beverages	Carbonated Beverages	Tea Beverages	Beverages for Special Uses
Total		5.3 (1.0–15.2)	0 (0–2.5)	0.8 (0–3.8)	0 (0–1.7)	0 (0–0.9)	0 (0)
Gender	Boys	6.0 (1.2–16.3)	0 (0–2.1)	0.7 (0–3.8)	0 (0–2.3)	0 (0–0.4)	0 (0)
	Girls	4.8 (0.8–14.3)	0 (0–2.6)	0.8 (0–3.8)	0 (0–0.6)	0 (0–0.9)	0 (0)
*p*		0.039	0.058	0.01	<0.001	0.259	<0.001
Age	6–8	1.8 (0–5.8)	0 (0–1.4)	0.5 (0–2.1)	0(0)	0 (0)	0 (0)
	8–10	6.0 (1.1–16.4)	0 (0–3.2)	0.8 (0–4.4)	0 (0–1.7)	0 (0–0.6)	0 (1.2)
	10–12	7.4 (2.4–19.1)	0 (0–3.5)	0.9 (0–4.2)	0 (0–2.9)	0 (0–2.1)	0 (0–0.2)
	12–14	8.5 (2.1–22.2)	0 (0–2.8)	0 (0.9–4.2)	0 (0–3.6)	0 (0–3.8)	0 (0)
*p*		<0.001	0.01	0.033	<0.001	<0.001	<0.001
Residence	Urban	3.6 (0.4–10.9)	0 (0–2.1)	0.4 (0–2.5)	0(0)	0 (0)	0 (0)
	Rural	6.9 (1.3–18.3)	0 (0–2.8)	1.0 (0–4.2)	0 (0–2.5)	0 (0–1.7)	0 (0–0.4)
*p*		<0.001	<0.001	<0.001	<0.001	<0.001	<0.001

**Table 5 nutrients-13-04297-t005:** Differences in sugar consumption level between different kinds of students (*n*, %).

Categories		Low- Consumption Group	Middle- Consumption Group	High- Consumption Group	*p*
Total		3325 (85.1)	352 (9.0)	231 (5.9)	
Gender	Boys	1648 (83.6)	199 (10.1)	124 (6.3)	0.027
	Girls	1677 (86.6)	153 (7.9)	107 (5.5)	
Age	6–8	984 (97.4)	18 (1.8)	8 (0.8)	<0.001
	8–10	819 (84.1)	88 (9.0)	67 (6.9)	
	10–12	753 (80.4)	114 (12.2)	70 (7.5)	
	12–14	767 (77.9)	132 (13.4)	86 (8.7)	
Residence	Urban	1469 (90.3)	95 (5.8)	63 (3.9)	<0.001
	Rural	1856 (81.4)	257 (11.2)	168 (7.4)	

**Table 6 nutrients-13-04297-t006:** Ordinal logistic regression of association between sugar consumption level and demography statistic among students.

Variable	B	SE	*p*	OR	95% CI	
Threshold						
Low-consumption level	−3.163	0.427	<0.001	-	-	-
Middle-consumption level	−2.089	0.429	<0.001	-	-	-
Gender (ref: girls)	0.269	0.0927	0.004	1.309	1.091	1.570
Age (ref: aged 12–14)						
Aged 6–8	−2.439	0.2139	<0.001	0.087	0.057	0.133
Aged 8–10	−0.438	0.1171	<0.001	0.645	0.513	0.811
Aged 10–12	−0.192	0.1132	0.09	0.825	0.661	1.030
Residence (ref: rural)	−0.800	0.1013	<0.001	0.449	0.368	0.548

## Data Availability

The data presented in this study are available on request from the corresponding author. The data are not publicly available due to privacy.

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
