# Peer review of "Sugar Content of Market Beverages and Children’s Sugar Intake from Beverages in Beijing, China"

_nutrients, 2021, doi:10.3390/nu13124297_

Round 1

Reviewer 1 Report

  1. Please, be more specific about the enrollment of students. Were they approached directly or through their parents? Did participants have to give written informed consent.
  2. Given the large number of participants, I am a little bit surprised that the authors created only 2 age groups. Would it be an idea to create more age groups?
  3. Please, remove phrasing from the Results section that has nothing to do with the actual results; eg, "A daily intake of at most 50 g added sugar was required in the Dietary Guideline for 258 Chinese Residents, and best not exceeding 25 g per day."

Author Response

Dear reviewer,

We gratefully thank you for your careful reading, positive and helpful comments, and constructive suggestions, which has significantly improved the quality of our manuscript. Below the comments are response point by point and the revisions are indicated.

1. Comment: Please, be more specific about the enrollment of students. Were they approached directly or through their parents? Did participants have to give written informed consent.

1. Reply: We gratefully appreciate for your comment. The students were enrolled directly in school by the investigators. All the participants have been given the written informed consent. Thank you again for your valuable comment. More specific enrollment of students has been added in the revised manuscript and shown below as well (line 136~140 in the revised manuscript).

Students were recruited directly in school by investigators. The survey was conducted via questionnaire in all recruited students and adults to investigate the consumption of snacks and beverages of students and the related family environment. Written informed consent was obtained from the students and their parents and all information was kept confidential.

2. Comment: Given the large number of participants, I am a little bit surprised that the authors created only 2 age groups. Would it be an idea to create more age groups?

2. Reply: Thanks for your comments, which is highly appreciated. According to your suggestion, we have modified our age groups from the original 2 groups to 4 groups with a 2-year-old interval. We have “6~8 years old”, “8~10 years-old”, “10~12 years old” and “12~14 years old” age groups in our revised manuscript and we also re-analyze the data by the new age groups. Thanks again for your valuable comment.

3. Comment: Please, remove phrasing from the Results section that has nothing to do with the actual results; eg, "A daily intake of at most 50 g added sugar was required in the Dietary Guideline for 258 Chinese Residents, and best not exceeding 25 g per day."

3. Reply: We appreciate for your valuable comment. Thank you so much for your careful check. We have deleted the irrelevant words and we feel sorry for our carelessness.

Many grammatical or typographical errors have been revised. All the lines and pages indicated above are in the revised manuscript. We sincerely hope that this revised manuscript has addressed all your comments and suggestions. We appreciated for your warm work earnestly, and hope that the correction will meet with approval.

Once again, thank you very much for your comments and suggestions.

Sincerely yours,

Jing Wen

Reviewer 2 Report

Sugar content of market beverages and children’s sugar intake 2 from beverages in Beijing, China

Jin Wen et al aimed to investigate the beverages consumption among students and how much sugar students could get daily from beverages.

The study is very interesting as it permits to take a sight into the Chinese’s alimentary behavior.

I think the paper is well structured and clearly states materials and methods and results.

However, authors have not full exploited the data.

Major Issues

  • The paper needs an intensive editing by an English native speaker
  • Please, state a clear objective by joining the two parts of this paper
  • Authors have only described the data by showing the number and type of beverages (and their sugar contents) and participants consumption rate and intake of sugar. These data have been described with the ethods stated in Statistical Analysis section
  • There is no inference in this paper and this could be useful to prompt to a generalization of the results. By performing a logistic regression this task may be accomplished easily

Please consider the opportunity to re-analyze the data. This may grant the paper an other level

Author Response

Dear reviewer,

    We gratefully thank you for your careful reading, positive and helpful comments, and constructive suggestions, which has significantly improved the quality of our manuscript. Below the comments are response point by point and the revisions are indicated.

1. Comment: The paper needs an intensive editing by an English native speaker

1. Response: We are very sorry for the mistakes in this manuscript and inconvenience they caused in your reading. The manuscript has been thoroughly revised, polished with the help of editing service and also marked out in the revised manuscript, so we hope it can meet the standard. Thanks so much for your useful comments.

2. Comment: Please, state a clear objective by joining the two parts of this paper

2. Response: Thank you for your comment. We have deleted some redundant descriptions and revised our statement of objective. The revised version was as below (line 68~74 in the revised manuscript):

Thus, in order to investigate the sugar content of market beverages and how much sugar students could get daily from beverages, we combined the statistic of a cross-sectional study with a consumption survey. A cross-sectional study with a sample of 26 supermarkets in Beijing was performed from October to December 2019 to investigate the sugar content of beverages. Besides, aiming to investigate the beverages consumption among students in Beijing, we also selected a portion of the data from the 2019 Students Snacks and Beverages Consumption Survey in Beijing for a secondary data analysis.

3. Comment: Authors have only described the data by showing the number and type of beverages (and their sugar contents) and participants consumption rate and intake of sugar. These data have been described with the methods stated in Statistical Analysis section.

There is no inference in this paper and this could be useful to prompt to a generalization of the results. By performing a logistic regression this task may be accomplished easily.

Please consider the opportunity to re-analyze the data. This may grant the paper another level.

3. Response: Thank you very much for the positive comments and constructive suggestions. We are sorry that the data description methods were not clear in the original manuscript. I should have explained that we introduced our collection methods of data on sugar content of beverage and our selection and conversion methods of data on student beverage consumption in the Materials and Method section in our original manuscript version. In our original manuscript, we introduced the statistical analysis methods among all kinds of variables in the Statistical Analysis section and we displayed the result in our Result section. These parts complemented each other and were not duplicated and we have deleted some redundant descriptions in our revised manuscript.

    In our revised manuscript, we have modified the age groups from the original 2 groups to 4 groups with a 2-year-old interval and we have re-analyzed the data. According to your suggestions, we have performed an ordinal logistic regression in our revised manuscript to explore the factors affecting the sugar consumption level among students. In addition, we have added some discussion in our revised manuscript. We sincerely hope that this revised manuscript has addressed all your comments and suggestions. Thank you again for your valuable suggestions.

    Many grammatical or typographical errors have been revised. All the lines and pages indicated above are in the revised manuscript. We sincerely hope that this revised manuscript has addressed all your comments and suggestions. We appreciated for your warm work earnestly, and hope that the correction will meet with approval.

Once again, thank you very much for your comments and suggestions.

Sincerely yours,

Jing Wen

Reviewer 3 Report

This research article is interesting since is the first to investigate the sugar content of beverages and children's sugar intake from beverages in Beijing, China. 

One comment.

Material and Methods

line 124-125. Make it more clear and write it better. You write sugar content per 100 ml or for portion. What portion? The same portion? For example here: low<= 2.5g. 2 you cannot understand. 

Author Response

Dear reviewer,

We gratefully thank you for your careful reading, positive and helpful comments, and constructive suggestions, which has significantly improved the quality of our manuscript. Below the comments are response point by point and the revisions are indicated.

1. Comment: Material and Methods

line 124-125. Make it more clear and write it better. You write sugar content per 100 ml or for portion. What portion? The same portion? For example here: low<= 2.5g. 2 you cannot understand.

1. Response: Thank you for pointing out this problem in our manuscript. According to the FSA criteria in the UK, the cuts-off of high sugar content are divided into 2 situations. For per 100ml, the cut-off of high sugar content beverages is > 11.5g. For per portion, the cut-off of high sugar content in beverages is >13.5g. We have corrected the expression as below (line 121~123 in the revised manuscript):

Cut-offs for the sugar content criteria per 100 ml in this guideline were as follows: high >11.25 g, medium >2.5 g and ≤11.25 g, low ≤2.5 g. Cut-off of high sugar content criterion per portion was >13.5 g.

We sincerely hope that this revised manuscript has addressed all your comments and suggestions. We appreciated for your warm work earnestly, and hope that the correction will meet with approval.

Once again, thank you for the kind advice.

Best regards,

Jing Wen